# Endoscopic Management of Postoperative Esophageal and Upper GI Defects—A Narrative Review

**DOI:** 10.3390/medicina59010136

**Published:** 2023-01-10

**Authors:** Cecilia Binda, Carlo Felix Maria Jung, Stefano Fabbri, Paolo Giuffrida, Monica Sbrancia, Chiara Coluccio, Giulia Gibiino, Carlo Fabbri

**Affiliations:** 1Gastroenterology and Digestive Endoscopy Unit, Forli—Cesena Hospitals, AUSL Romagna, 47121 Forlì Cesena, Italy; 2Department of Health Promotion, Mother and Child Care, Internal Medicine and Medical Specialties, PROMISE, University of Palermo, 90127 Palermo, Italy

**Keywords:** endoscopic treatment of anastomotic defects, esophageal fistula, perforation, esophageal leakage

## Abstract

Anastomotic defects are deleterious complications after either oncologic or bariatric surgery, leading to high morbidity and mortality. Besides surgical revision in early stages or instable patients, endoscopic treatment has become the mainstay. To date, many options for endoscopic treatment in this setting exist, including fully covered metal stent placement, endoscopic vacuum therapy (EVT), endoscopic internal drainage with pigtail placement (EID), leak closure with through the scope or over the scope clips, endoluminal suturing, fibrin glue sealing and a combination of all these techniques. Current evidence is mostly based on retrospective single and multicenter studies. No guidelines exist in this important field. Treatment options have to be chosen upon each case individually, taking into account clinical and anatomic criteria, such as timing, size, infectious wound complications and hemodynamic stability. Local expertise and availability of treatment devices need to be taken into account whenever choosing a treatment strategy. This review aimed to present current treatment options in terms of effectiveness, advantages and disadvantages in order to guide the clinician for his decision making. Additionally, we aimed to provide a treatment algorithm.

## 1. Introduction

Leaks, fistulas and anastomotic defects after either oncologic or bariatric surgery are feared complications. They occur in up to 13.1% of cases after esophagectomy [1,2] and in up to 7.5% after gastrectomy in large patient cohorts [3,4]. Leakage sites may be intrathoracic or intra-abdominal in these patients.

Patients undergoing bariatric surgery including sleeve gastrectomy and Roux en Y bypass may also be subject to postoperative anastomotic defects, leakages or fistulas. After gastric sleeve, they are reported in 1–3.9% [5,6,7] and after Roux en Y bypass, in 0.6–5.25% [8,9,10]. Leak site after sleeve gastrectomy typically is situated at the stapler line whereas after bypass surgery the leak most commonly (in up to 50%) is situated at the gastrojejunal anastomosis followed by the gastric pouch [11,12].

Classifications try to systemize clinical appearance and treatment decisions.

According to the ECCG (esophagectomy complications consensus group), postoperative defects after esophageal surgery are defined as full thickness GI defects involving the esophagus, anastomosis, staple line or conduit. They are subclassified into three groups: type I local defect requiring no change in therapy; type II localized defect requiring interventional but not surgical therapy and type III localized defect requiring surgical therapy [13]. Defects can also be classified by timing of occurrence, for which either surgical reintervention or endoscopic management might be the treatment of choice. Here, a general accepted subdivision at least for esophageal and post-gastrectomy defects is proposed by Bludau et al.

“Early leaks “are considered to occur within 72 h after operation, “classic leaks are observed in a period from 4–10 days after operation and late leaks occur after postoperative day 10 [14].

Besides conservative treatment, septic conditions frequently require surgical re-intervention or application of endoscopic therapies.

For anastomotic defects after oncologic upper GI surgery (Ivor Lewis and gastrectomy), early surgical reintervention in these conditions is favored during the first postoperative days. Endoscopic therapies remain treatment of choice after the third postoperative days [14,15,16,17].

After bariatric operations: surgical reintervention will normally be performed in early leaks (<5 days postop) whereas non-surgical treatment is favored in patients with chronic leaks [18].

Many options for the endoscopic treatment for any kind of anastomotic defect exist, including fully covered metal stent placement, endoscopic vacuum therapy (EVT), endoscopic internal drainage with pigtail placement (EID), leak closure with through the scope or over the scope clips, endoluminal suturing, fibrin glue sealing and a combination. Table 1 summarizes the main features of endoscopic closure techniques of upper GI fistula, leaks and perforations.

Until now, no guidelines for endoscopic treatment for defects in the upper GI after oncologic or bariatric surgery are available. Current ESGE guidelines only comprise treatment suggestions for intestinal perforations (for any cause) [19]. Large quantitative or qualitative data which could serve for guidelines are still scarce. The data published mostly consist of retrospective single and multicenter studies. Most definitely, due to ethical aspects, randomized controlled trials will hardly exist.

No leak/fistula or defect is anatomically the same and needs individual treatment approaches, which might be changed during the course of treatment. Time of detection, anatomy, size, presence of a wound cavity including presence of a drainage need to be taken into account in order to choose the right endoscopic armamentarium.

In this narrative review, we look at the current literature comprising endoscopic techniques for the management of upper GI anastomotic defects, their indications, contraindications, technical aspects, treatment algorithms and complications.

## 2. Endoscopic Techniques

### 2.1. Stenting

Endoluminal stent placement has been proven to be a safe and effective treatment option for upper GI leaks and fistula. Once released under fluoroscopic or endoscopic control or both, the meshes expand radially to its maximal diameter and the stent adheres to the mucosal wall. The rationale of stent deployment is to seal the breach and divert luminal content, allowing the closure of wall defect. Diversion therapy offers the advantages of early oral intake and early discharge. In addition, stent placement prevents the onset of gastric stenosis in patients affected by sleeve gastrectomy leak [20].

In two recent published guidelines, ESGE recommends that temporary stent placement can be considered for the treatment of leaks, fistulas and perforations >2 cm in size, but no specific type of stent can be recommended [19,21]. Indeed, different types of stents are commercially available: self-expandable plastic stents (SEPS), self-expandable metal stents (SEMS), both fully covered (FCSEMS) or partially covered (PCSEMS) and biodegradable stents.

The most used self-expandable plastic stent (SEPS) is Polyflex (Boston Scientific, Natick, MA, USA), made of polyester, completely covered with silicon. SEPSs guarantee an easy removal and a low cost, but their use is burdened by a high rate of migration [22].

Self-expandable metal stents (SEMSs) are composed of various metal alloys, which confer them a higher radial force compared to SEPSs. FCSEMSs have a plastic or silicone rubber coating along its full length. PCSEMSs has uncovered distal and proximal ends. These help for the optimal fitting and prevention of migration. On the other hand, uncovered parts of the stent are exposed to mucosal in-growth. This could lead to increased risk of bleeding, mucosal stripping and perforation during stent removal. A recent multicenter retrospective study showed that FCSEMSs were more successfully removed than self-expandable plastic stents and PCSEMSs. However, esophageal stent removal in the setting of benign disease was affected by a low rate of adverse events (2.1%) [23].

The results of three systematic reviews on the use of PCSEMSs, FCSEMSs, and SEPSs reported a clinical success rate of esophageal stent placement of 81%–87%, with similar efficacy between the different stents (SEPS 84%; FCSEMS 85%; PCSEMS 86%; *p* = 0.97) [24,25,26]. SEMSs are reported to perform better than SEPSs in leaks and perforations, with higher technical success (95% vs. 91%; *p* = 0.03), and reduced risk of migration (16% vs. 24%; *p* = 0.001) and stent repositioning (3% vs. 11%; *p* < 0.001) [26].

Freeman et al. identified four factors associated with failure of stenting therapy for esophageal wall defects, such as location of the defect at the proximal cervical esophagus, stent traversing the gastroesophageal junction, esophageal injury longer than 6 cm and an anastomotic leak associated with a more distal conduit leak [27]. The presence of fluid collection is another unfavorable factor in successful treatment with stents, especially when the fluid collection is >5 cm, thus appropriate drainage of any pre-existing or concurrent extra-luminal collection is mandatory [28].

The most frequently reported adverse event is stent migration, which is higher for FCSEMSs (26%) and SEPSs (31%) compared with PCSEMSs (12%), as might be expected [24]. Migration risk can be reduced by fixating the proximal flange of the stent to the esophageal wall with TTSC, OTSC or endosuturing devices. Ngamruengphong et al. found no statistically significant difference in stent migration rate between PCSEMS and FCSEMS fixed with the OverStitch suturing device (Apollo Endosurgery, Austin, TX, United States) [29]. Moreover, fixation of FCSEMSs with a novel dedicated over-the-scope clip device, Stentfix OTSC^®^ (Ovesco Endoscopy, Tubingen, Germany) significantly reduced migration rate compared with unfixed stents [30].

Other stent-related adverse events include stricture development, stent rupture, food impaction, mucosal erosion with perforation or massive bleeding due to erosion into the major vessel [24,25,26,31].

According to the ESGE guidelines, the timing of stent retrieval is still subject to debate [21]. Stents are usually removed 6–8 weeks after insertion [24,25,26], but some authors report success with only 2 weeks of deployment [32]. In order to avoid complications, Van Heel et al. suggests stent indwelling within 6 weeks after insertion [33]. A survey questionnaire, distributed among international expert interventional endoscopists, reported the tendency to reduce the stent dwell time to 4–5 weeks in clinical practice [34].

Biodegradable stents (BDs) such as SX-Ella (Milady Horakrove, Hradc Kralove, Czech Republic) are made of polydioxanone, an absorbable polymer which degrades after 3 to 4 months in a low ambient pH. Therefore, it is a potentially ideal solution for temporary use in benign indications because BDs do not need to be removed, but data regarding their use in clinical practice are still limited. Promising results were published by Cerna et al [35] who reported a case series of five patients with esophageal perforation or anastomotic leak treated with covered biodegradable stents. Technical success was achieved in 100% of patients and clinical success was achieved in four out of five patients (80%), but stent migration occurred in three patients (60%). Although biodegradable stents eliminate complications involved in stent removal, they are more expensive. Additional side effects reported in the literature include drooling, retrosternal pain, “foreign body”—sensation and aversion to water for up to 2 months [36].

Customized SEMS have been recently designed for the treatment of leaks after bariatric surgery, especially for sleeve gastrectomy leaks (SGL). Customized bariatric stents (CBS) include Niti-S Mega [37,38] and Niti-S Beta [39,40] (TaeWoong Medical Industries, Seoul, Republic of Korea), Hanaro GastroSeal [41] and Hanaro ECBB [42] (M.I. Tech, Seoul, Republic of Korea). These stents have common characteristics: CBS are fully covered SEMSs with a longer length (18–24 cm) that ensures the complete coverage of the leak area and to bypass the wide gastric lumen. A large diameter ensures a complete seal and reduces the risk of migration. Significant flexibility allows to conform to the tortuous bariatric surgery anatomy. Moreover, every CBS is equipped with a specific anti-migration system.

Hamid et al. has recently published a systematic review and meta-analysis aiming to evaluate the cumulative efficacy and safety of CBS and to compare them with the conventional esophageal stents (CES) [43]. In total, 12 studies (141 patients) used CBS and 11 studies (167 patients) used CES. Treatment with CBS was associated with a similar technical success rate, fewer stent insertions and endoscopic interventions, and shorter time to leak closure compared to CES. Non-Niti-S Mega stents had a higher clinical success rate (89%) than Niti-S Mega stents (66%), and a similar clinical success rate to CES (93%). Of note, in the Niti-S Mega group, 11% of patients had combined sleeve leakage and stenosis, which were reported to be associated with lower clinical success [7]. On the other hand, non-Niti-S Mega stents had the highest migration rate (41%) compared to other types of stents, including CES and Niti-S Mega stents (15–24%). Anyway, the overall quality of evidence was very low and further studies, including randomized trials, are warranted.

### 2.2. Endoscopic Clip Placement (TTSC and OTSC)

Through-the-scope clips (TTSC) are available in different sizes and opening lengths. TTSC are inserted and deployed through the operative channel. Multiple clips can be applied in a parallel manner.

A recent updated ESGE position paper suggests the use of TTSC for upper GI leaks or perforations <1 cm in size, in general due to small clip size and low tissue compression force [19]. Their application is also limited by location of the defect and endoscopist experience [44,45].

Clip application might be difficult and lead to suboptimal closure if the tissue surrounding the defect is inflamed, necrotic or fibrotic.

A pooled analysis performed by Qadeer et al. demonstrated that TTSC can be effective for closing both acute and chronic esophageal defects; however, there is a statistically significant correlation between the duration of a perforation and the time of healing, which is longer for chronic perforations than for acute perforations [46].

A retrospective study including 20 patients with anastomotic leak after gastric surgery, published by Lee et al., reported a 95% technical and 100% clinical success rate after TTS clip deployment [47].

Other than the TTSC, over-the-scope clips have demonstrated several advantages in closing GI defects, including as the ability to capture larger area of tissue and applying higher compression force. OTS clip can achieve full-thickness closure of GI defects up to 2–3 cm. In the recent published ESGE Position Paper, the use of OTSC is suggested for upper GI leaks or perforations larger than 1 cm in size [19].

The first developed and most used OTS clip is the OTSC^®^ (Ovesco Endoscopy AG, Tübingen, Germany). The OTSC^®^ has a bear-trap shape design made of Nitinol, a biocompatible, super-elastic, shape-memory material which firmly anchors the tissue and can remain in the body as a long-term implant.

The edges of the wall defect can be approximated and pulled inside the application cap mounted at the tip of the endoscope by simply applying suction. Otherwise, the edges can be pulled actively into the cap using additional devices (twin grasper/tissue anchor).

Once the whole leak is engulfed into the cap, the clip is deployed by turning an handwheel located in the endoscope handle, similar to a variceal band ligator. OTSC^®^ are available in different sizes and with different dental shapes (atraumatic/traumatic).

The traumatic Ovesco OTSC, equipped with spiked teeth, is the most used to close fistula and perforations.

In a recent systematic review analyzing 381 patients with anastomotic leak, the overall technical and clinical success rate for OTSCs closure was 86.7 and 72.6%, respectively [48]. A large single center case series included in this review reported long-term clinical resolution in 83% patients affected by post-operative leaks in the upper GI-tract [49].

In 2021, Rogalski et al. published a systematic review and meta-analysis of 13 studies including a total of 85 cases of leaks and fistulas after bariatric surgery [50]. Overall, successful closure of a leak/fistula with the OTSC system was achieved in 57 of 85 patients (67.1%). Only two studies reported complications related to the OTSC system including clip migration (1 patient), mediogastric stenosis (1 patient) and one case of anchor blocked within the clip during the deployment.

Another advantage of OTS over TTS clips is the ability to close long-term leakages and fistulas even if the surrounding tissue is inflamed or fibrotic. Several studies suggest to de-epithelialize the edges of the fistula with Argon Plasma Coagulation or with a cytology brush before OTS clip placement in order to promote granulation tissue and obtain a stronger grip of the tissues [20].

A large multicenter retrospective study was published in 2014 by Haito-Chavez et al. including 188 patients undergoing OTSC placement for closure of GI defects, of which 62.8% were in the upper GI tract. The rate for the successful closure of perforations (90%) and leaks (73.3%) was significantly higher than that of the fistulae (42.9%) (*p* < 0.05). Long-term clinical success did not differ between all three defect types but was significantly higher when OTSCs were applied as primary therapy as compared with rescue therapy (69.1% vs. 46.9%, respectively; *p* = 0.004) [51].

If the fistula is in communication with an abscess cavity, OTSC is significantly more efficient in the case of patients having a prior endoluminal drainage (88.2% healing in this subgroup vs. 53.8%, *p* = 0.049) [52].

In conclusion, the use of OTS clips is suggested in the case of early detection, perforation diameter ranging from 10 to 20 mm and the absence of fluid collections [45].

A new OTSC device, the Padlock Clip (Aponos Medical, Kingston, NH, United States), has been introduced [53]. It differs from Ovesco in the hexagonal shape and in the deployment system. Clinical data are currently limited, but several case reports describe the successful treatment of tracheoesophageal fistula [54], gastrocutaneous fistula [55] and iatrogenic duodenal perforation [56].

### 2.3. Endoscopic Suturing

Recent development of endoscopic suturing techniques has allowed for the full-thickness closure of large GI luminal defects. The OverStitch system (Apollo Endosurgery, TX, USA) was first developed in 2009 and is currently the most common endoscopic suturing device [57]. The original OverStitch is a single operator, disposable platform that requires a double-channel therapeutic endoscope. Instead, the newly introduced OverStitch SX (Apollo Endosurgery, TX, United States) can be mounted on every single channel endoscope commercially available. The device is composed of a handle attached to the endoscope controls, a metallic needle on the tip of the endoscope, devices for tissue retraction and a specially designed non absorbable suture.

Endoscopic suturing is a complex technique, requiring specific training and a high level of expertise, limiting its use to tertiary centers only. Due to its size and reduced maneuverability, endoscopic suturing may be challenging in narrow or angulated GI locations, such as the gastric fundus, duodenum or sigmoid colon. Similar to TTS clips, suturing requires robust and healthy mucosa to hold the sutures when tissues are approximated and is therefore adapted for early leakages in the absence of an associated wound cavity.

In a large multicenter retrospective study, including 122 patients undergoing endoscopic suturing, clinical success was 91.4% in stent anchorage, 93% in perforations, 80% in fistulas, but only 27% in anastomotic leak closure [58]. Long-term clinical success was more likely if the leak was closed within days of diagnosis, indicating its usefulness in mainly treating acute and early leaks.

Chon et al. performed a retrospective, single-center study of 13 patients affected by leaks in the upper gastrointestinal tract treated with OverStitch [59]. The mean size of the leak was 22.31 ± 22.6 mm. Interventional success was achieved in all endoscopic attempts (*n* = 16, 100%) with a mean closure time of 28.0 ± 12.36 min per patient. Clinical success was achieved in 8 of the 13 patients (61.5%). These eight patients had not received prior treatment for the leak.

A recently published systematic review and meta-analysis showed a pooled technical success for any GI defect treated with Apollo OverStitch of 92.7% (95% [84.4–96.8]), clinical success was 67.9% [59.2–75.5], and adverse events occurred in 6.9% [3.8–12.5] [60]. The pooled clinical success for perforations was higher when compared to fistulae/leaks (respectively 89.5% [73.8–96.3] vs. 60.4% [50.1–69.9]).

Endoscopic suturing can also be used for esophageal stent fixation in order to prevent migration [29]. In this regard, Granata et al. reported a recent case series of 20 patients with post-operative leaks [61]. The therapeutic approach was stratified in three groups according to the clinical scenario and structural condition of the wall defect layers: Pure endoscopic direct suture (group A: healthy tissue and feasible suture), combined therapy with endoscopic direct suture + FC-SEMS placement + anchoring (group B: unhealthy tissue and feasible sutures) and FC-SEMS placement + anchoring (group C: unhealthy tissue and suture not feasible). The overall long-term clinical success was 80%. The clinical success rate for each group was 77% (7/9) in group A, 85% (6/7) in group B and 75% (3/4) in group C. No evidence of migration was detected.

Endoscopic suturing is a well-accepted treatment option for long-term complications after bariatric surgery such as dilation of the gastrojejunal anastomosis (TORE procedure) [62]. There are few studies explicitly examining the effectiveness of endoscopic suturing of anastomotic malformations and stapler line leaks after bariatric surgery. Therefore, no conclusions or recommendations can be drawn for this indication.

### 2.4. Endoscopic Vacuum Therapy (EVT)

Endoscopic vacuum therapy has been applied for the treatment of various types of defects in the gastrointestinal tract. First applied for anastomotic leakage after colonic surgery with promising results, it was successfully used for the treatment of anastomotic defects after upper GI oncologic surgery [63,64,65,66].

So far, EVT has mostly been used for defects (leaks, fistulas, but also perforations) in the upper GI after oncologic surgery. Recent systematic reviews confirm high defect closure rates (81.6–85%) for these indications [67,68]. Less frequently EVT has also been proposed for the treatment of defects/leaks after bariatric surgery (staple line defects after laparoscopic sleeve gastrectomy and anastomotic defects after Roux en Y bypass) showing high healing rates up to 90% [69,70,71,72].

EVT consists of an open pore polyurethane sponge attached to a suction tube to which negative pressure of −125 mmHg is applied (for example Eso-Sponge^®^ Braun B Melsungen Germany); see also Figure 1. The sponge and suction tube are inserted endoscopically via an overtube, externalized transnasally and attached to a suction device which generates negative pressure. EVT works through at least five different mechanisms: 1. wound adaption; 2. cavitary collapse; 3. induction of angiogenesis; 4. wound granulation; and 5. bacterial clearance.

Animal experiments showed the highest efficiency of negative pressure therapy for inducing granulation, wound adaption and induction of angiogenesis at around −125 mmHg [73]. Studies with negative pressure inferior to −125 mmHg exist but are scarce. Here, EVT is considered to act by diverting possibly toxic fluids (bile, pancreatic enzymes, gastric acid) away from the anastomosis, and less through wound adaption/cavity collapsion and aspiration of bacterial fluids. Nevertheless, administration of low negative pressure around −50 mmHg still proved to be efficient [74]. Loske et al. recently described postoperative pre-emptive active drainage of reflux using negative pressure with a 25 cm open pore film drainage esosponge device after Ivor Lewis esophagectomy to be effective for anastomoses at risk with focal necrosis [75]. No randomized controlled trials investigating the effect on different negative pressures exist.

Lately, a new form of EVT using an open pore film wrapped around the drainage tube (Suprasorb CNP, Drainage Film; Lohmann & Rauscher International GmbH & Co. KG, Rengsdorf, Germany) has been proposed by Loske et al. Its advantage lies in its capability of being applied to cavities behind small defects (4–6 mm), where a classic polyurethan sponge cannot be placed due to its diameter (1.5–3.2 cm) [76].

EVT can be applied either intracavitarily or intraluminally and is adapted for different sizes of defects. When defects measure <10 mm, EVT is generally placed intraluminally outside of the wound cavity. In defects >10 mm, EVT can be placed intracavitarily. In patients with complete anastomotic dehiscence after for example Ivor Lewis operation, EVT can still be placed, although it may not represent the ideal indication [15]. Defects with a size of less than 10 mm can sometimes be dilated in order to fit in the Eso-Sponge^®^ system.

In patients with large defects, EVT can be initiated intracavitarily and finished intraluminally. In case of two different defects, simultaneous intraluminal and intracavitary application can be performed [77].

EVT can come with complications. In the literature, procedure-related bleeding, sponge ingrowth and stricture development are described.

Minor bleeding can be addressed endoscopically, in general when it occurs directly associated with the sponge change. Only a few bleedings associated with sponge change have been described that led to fatal consequences. Laukoetter et al. described a case of aortic rupture into the esophagus not manageable by endoscopic means [17].

Strictures associated with therapy are rare (ca. 7.6%) and can be managed by endoscopic pneumatic dilation [17,78]. Recently, stricture incidences up to 35% have been presented in a smaller Korean study. This result must be interpreted carefully due to the small cohort size [79].

Sponge ingrowth poses a serious issue when the sponge is left in place for over 4 days [80]. Then, difficulties in removing may occur resulting in detachment of the sponge and the suction tube. Then, careful endoscopic resection of the ingrown sponge part is necessary.

#### Timing of EVT after Upper GI Surgery

This subject has not been addressed in major studies or meta-analysis so far. It is not clear whether late onset of therapy after diagnosis leads to longer treatment duration. Risk factors for long treatment so far are patients with neoadjuvant treatment and larger defect sizes >2 cm [79]. Another recent retrospective multicenter study by Hyun Jung et al. investigates factors associated with treatment failure in patients with leaks and perforations. Here, neoadjuvant treatment and interestingly, the intraluminal method, are independent risk factors for treatment failure [81].

According to our own experience, it is important to initiate therapy early after defect diagnosis. At this point, mucosal injury has not turned into fistulous or fibrotic tissue which generally makes it more difficult for EVT to induce wound granulation. Otherwise, risk of longer treatment duration is high. According to Bludau et al., EVT can generally be initiated 3 days after defect diagnosis. Prior to this point, early surgical revision is recommended [14].

Instead, some studies have looked into the prophylactic use of EVT in patients with at-risk anastomoses during upper GI surgery for those with high-risk comorbidities or anastomosis ischemia [82,83]. In the study by Laukoetter et al., high “protection” rates with a low incidence of defect development in anastomotic ischemia were observed (75% of patients). When defects occurred (25% of patients treated pre-emptively), EVT was continued until closure.

So far, no consensus was found to define the failing of EVT treatment. Treatment failure is still a clinical decision comprising leak persistence and ongoing purulent secretion.

Whether SEMS placement or EVT should be preferred for patients with upper GI defects after oncologic surgery is not clear.

By now there are no randomized controlled trials available comparing EVT treatment to SEMS placement. Only retrospective studies on this subject exist. Two recent review and metanalysis studies address this important comparison as these two therapies are most frequently used for the treatment of upper GI defects [84,85]. Here, EVT seems to be superior to SEMS therapy in terms of defect closure, mortality, hospital stay and adverse events. A first phase 2 trial (ESOLEAK-Trial) by Tachezy et al. will try to shed some light on this matter although the primary endpoint is “quality of life”. Group sizes in this protocol will not contain more than 20 patients per group (EVT vs. SEMS) [86].

Additionally, patients with defects after bariatric surgery (in case of staple line leaks) seem to profit from EVT compared to SEMS placement. In a study by Archid et al. in 24 patients either treated with EVT or SEMS for staple line leakage after sleeve gastrectomy, EVT was shown to be superior in terms of defect closure, reducing adverse events, hospital stay and duration of endoscopic treatment [87].

More studies using EVT for defects after bariatric surgery are needed.

### 2.5. Endoscopic Internal Drainage (EID)

Endoscopic internal drainage proves to be a valuable alternative to SEMS—and EVT placement in the treatment of anastomotic defects. First introduced by Pequignot et al. in 2012 and later by Donatelli et al., it was used for defects after bariatric and oncologic esophageal surgery showing promising results [88,89,90,91,92]. From there, many other single center studies showed that EID provides high healing rates up to 78–95% for defects after either upper GI oncologic interventions or bariatric operations [92,93,94,95,96]. The largest single center study by Donatelli et al. describes an experience of 617 EID cases for defects following bariatric surgery demonstrating a cumulative efficacy of 84.7% [97].

Endoscopic internal drainage consists of endoscopically placing one or multiple double pigtail plastic stents into the anastomotic defect and its associated cavity (see Figure 2). When suspecting anastomotic defects, the local situation is being examined by normal gastroscopy. Via fluoroscopy, the cavitary size behind the defect is evaluated. Then, defects are intubated with a straight catheter and a guidewire over which pigtail are inserted (normal pigtail size: 7–10 Fr, 3–5 cm).

EID works by at least two mechanisms. One is the passive drainage of purulent material accumulated in the cavity behind the anastomotic defect, and the second is by continuously irrigating the fistulous tract inducing wound granulation. In the studies demonstrated so far, EID is changed every 3 weeks with no need for hospitalization during treatment. Oral alimentation is generally possible during treatment. Whenever pigtail exchange is performed, local wound situations are examined, and treatment modalities can be adapted when necessary. So far, no clear treatment algorithms exist but it seems obvious that EID most sufficiently might be working in patients with anastomotic defects up to 2 cm of size. In defects greater than 2 cm, also when placing multiple stents, pigtails can easily dislocate and therefore not induce wound granulation and clearance of purulent fluids. Therefore, EID has no indication in patients with complete anastomotic dehiscence. More studies investigating the success rates including complication rate of EID treatment are necessary. Especially, direct comparative studies between SEMS placement and EID or EVT and EID are needed to confirm treatment effects and define optimal treatment indications.

A study by Lorenzo et al. compared the effect of EID vs. direct clip closure of defects after sleeve gastrectomy in a cohort of 100 patients. Primary success rates of EID were 86% whereas clip closure had an efficacy of only 63% [28].

A recent review and meta-analysis by Laopemathong et al. compared the effectiveness of EVT and EID: EVT had slightly inferior healing success rates with 85.2% compared to EID with 91.6% when used as first line treatment in patients with post-bariatric leaks [98].

A study by Hallit et al. compared effectiveness of endoscopic treatment with either EID or SEMS placement in patients with oncologic upper GI surgery. Here, in 68 patients with either prior Ivor Lewis esophagectomy, tr-incisional esophagectomy or total gastrectomy, the healing success rate for patients with defects treated by SEMS placement was 77%, whereas the healing success rate in EID was 95% [94].

A study by Jung et al. Comparing EID vs. EVT in defects after oncologic surgery confirmed high healing success after EID with 100% vs. 85.2% after EVT [99].

Only a few complications with EID treatment are described. The most common are ulcerations, upper gastrointestinal symptoms, splenic hematoma, stenosis, bleeding, stent migration (into the spleen or peritoneum) and pneumoperitoneum [89,95,97].

Still, larger studies are needed to confirm the effect, advantages, disadvantages and possible contraindications for EID treatment.

## 3. New Concepts

One of the new treatment concepts is the sponge over stent method, combining wound fluid suction and defect coverage. It enables draining wound secretions of cavities connected to anastomotic defects, whilst covering the defect itself and enabling liquid/food oral intake.

Technically, either a specially conceived device (Vac Stent GI Microtech^TM^ Endoscopy, Micro-Tech Europe GmbH) or a manually constructed device can be used [100,101].

The MicroTech Vac Stent can be used in defects up to 30 mm and consists of a nitinol stent covered with a silicone membrane and a 10 mm thick sponge system fixed to the outer layer of the stent. Either way, a sponge will be attached to a fully covered metal stent and placed over the anastomotic defect and associated cavity.

Only limited data for this method are available so far. In the study by Valli et al., a total of 12 patients with upper GI wall defects were treated with the stent over sponge method, in 7 patients as a first line treatment with a success rate of 71.4% and in 5 patients as a second line treatment with a success rate of 80%. No severe adverse events were observed [101].

In a preliminary study using the Microtech stent by Lange et al., a total of three patients with different types of defects were treated: one patient presenting a leak after subtotal esophagectomy, one patient with acute Boerhaave syndrome and one patient with a full thickness defect caused by an anti-reflux device. In all patients, successful defect closure was obtained.

An interesting feature of the SOS device is the necessity of changing the stent only every 5 days as sponge ingrowth does not seem to occur as frequently as compared to classic EVT.

A very promising new technique is described by Nachira/Boskoski et al. for leaks that have failed prior classic endoscopic treatment or which may not be treated either endoscopically or surgically for various reasons (e.g., anatomical difficulties). This new innovative method was tested in five patients (two with upper and three with distal esophageal fistulas). The method consists of the submucosal injection/delivery of the stromal vascular fraction obtained by the mechanical emulsification of autologous adipose tissue. The fluid comprises mesenchymal stem cells and fragments of the extracellular matrix obtained by centrifugation of subcutaneous fat of the patient itself. The injection is performed submucosally in each quadrant around the defect until obliteration of the defect. Fistula closure was obtained in all five cases after 7 days of injection. Long-term follow-up after a median of 8 months showed persistent defect closure [102]. These data should be confirmed in larger prospective studies.

## 4. Treatment Logarithm Proposal

A flowchart for treatment choices according to the clinical situation is illustrated in Figure 3 (modified after Loske et al. and Di Leo et al. [15,103]).

There are several main parameters that need to be considered when choosing the right treatment for defects after upper GI surgery (either after oncologic or bariatric surgery). This includes timing, anatomical site, presence of an associated wound cavity and hemodynamic stability of the patient.

First, the timing of diagnosis is of crucial interest. Timing of defect detection is divided in “early” <3 d, “classic/intermediate” 4–10 d and “late” >10 d after either bariatric or oncologic surgery [14,104]. Some authors consider early defects in patients after oncologic and bariatric surgery a classic indication for surgical re-intervention [5,14]. Otherwise, the presence of an associated wound cavity is of crucial interest.

In case of an associated wound cavity and intermediate or late diagnosis, it depends whether a surgical drain is still present in the wound cavity. If so, drain removal can be performed enabling treatment with an endoscopic draining method (EVT/EID). Otherwise, SEMS placement can be performed whilst leaving the drainage in place. When no drain is in place, EVT can be placed intracavitarily when the defect size is equal or larger than 2 cm, otherwise the defect can be artificially amplified by dilation; 2 cm is normally the minimal size allowing for EVT—overtube insertion into the defect associated cavity. If the defect is smaller than 2 cm, either EVT can be placed intraluminally or EID placement can be performed.

In patients where the defect has no associated wound cavity and defect diagnosis is made in an early stage, direct defect closure either with TTSC (defect size up to 1 cm) or OTSC (defect size between 1–2 cm) can be performed. The sizes and recommendations for clip use are referred to in the latest ESGE recommendations for endoscopic treatment in gastrointestinal perforations [19]. In selected cases with intact mucosal tissue, endoscopic suturing could be an option for defect closure. If the defect is not amenable to clipping or endoscopic suturing, intraluminal EVT can be performed. TTSC/OTSC placement or endoscopic suturing should not be performed if local wound infection is suspected and no drainages are in place.

Complete anastomotic dehiscence usually cannot be treated with EID. In these cases, when diagnosed >3 days after surgery, a trial with SEMS placement or EVT can be performed. If no wound granulation is observed, surgical reintervention might be necessary.

In general, EVT needs to be changed every 3–4 days. For EID, no clear change intervals are defined. In most publications, EID exchange is performed every 3 weeks. SEMS are generally exchanged every 4 weeks.

## 5. Discussion

SEMS, EVT and EID are the main treatment options for upper GI defect closure. In 2013, Schniewind et al. retrospectively compared mortality in patients treated with surgical revision, EVT and SEMS placement for defects after upper GI oncologic surgery. Once adjusted for APACHE II score, patients treated with surgical revision and SEMS patients showed significantly higher mortality than patients treated with EVT [105]. This study did not mention closure rates nor documents the postoperative timepoint of treatment after defect diagnosis. Nevertheless, it shows a paradigm shift concerning treatment of postoperative defects towards endoscopic approaches.

For a long time, SEMS placement was the mainstay of endoscopic treatment. With the invention of EVT and clinical data showing slightly higher defect closure rates in single center studies, a shift in first line treatment towards EVT was observed. Do Monte Junior and Scognamiglio et al. addressed this important topic in their meta-analysis for treatment efficacy between SEMS and EVT therapy. They included five studies retrospectively comparing the efficacy of SEMS and EVT in patients with upper GI defects. They found a significant 21% increase in successful defect closure in patients treated with EVT compared to SEMS. Other observations were a significant 12% reduction in mortality for patients treated with EVT compared to SEMS, an average reduction of treatment duration by 14.22 days with EVT vs. SEMS and a 24% reduction in adverse events in patients treated with EVT vs. SEMS [84,85]. Obviously, EVT therapy was associated with higher number of endoscopic interventional sessions.

EID treatment seems to be a valid and less expensive alternative to EVT, needing fewer interventions, leading to high closure rates. Comparative analyses for EVT and EID are extremely scarce. Our literature research has evidenced one study comparing EVT vs. EID in defects after oncologic upper GI surgery with higher treatment success rates in EID treatment (100% overall treatment success in EID vs. 85.2% in EVT, *p* = 0.03) [99]. EID also seems to be superior to EVT for defect closure after bariatric surgery, as outlined in the meta-analysis be Laopemathong et al. (91.6% EID vs. 85.2% EVT) [98]. One recent multicenter retrospective analysis compared the treatment success of SEMS vs. EID in upper GI defects after oncologic surgery, concluding in favor of a higher treatment success after EID (95% EID vs. 77% SEMS, *p* = 0.06) [94].

The scarcity of data makes it difficult to draw final conclusions whether supporting SEMS, EVT or EID as the first line treatment for defect closure either after upper GI oncologic or bariatric surgery. Nevertheless, data published so far seem to favor EVT and EID over SEMS placement as first line therapy. Nonetheless, whenever one treatment technique does not show sufficient defect closure, treatment reevaluation needs to be performed, not excluding device changes. One point not being addressed in studies is local availability of treatment methods and the clinical experience of the endoscopist. Our recommendation is to treat patients with defects in tertiary clinical centers or where sufficient clinical endoscopic expertise in all main closure techniques and intensive care are provided.

Larger prospective, comparative multicenter studies are clearly needed in order to guide clinicians in their decision making. Upper GI defects remain very difficult to treat, not only due to individual anatomic properties of the defect but also due to individual patient comorbidities.

## 6. Conclusions

Upper GI anastomotic defects after either oncologic or bariatric surgery come along with high morbidity and mortality. Multiple endoscopic treatment modalities exist and they have to be applied looking at each case individually and upon multidisciplinary agreement. The mainstay of therapy has been SEMS placement for over a decade leading to healing success in up to 87% in these scenarios. Indications for SEMS placement range from defects under 1 cm up to complete anastomotic dehiscence. If purulent wound cavities are associated with defects, external drainage besides SEMS placement is necessary. TTSC and OTSC placement can be performed in defects from 5 mm up to ca. 2 cm in early diagnosed defects without wound cavities or as a final closure device when cavities behind defects have been cleaned. Endoscopic vacuum therapy provides high healing rates >90% and can be placed either intraluminally or intracavitarily. Defect size can range from 5 mm to complete anastomotic dehiscence. Endoscopic internal drainage shows high healing success rates up to 95% associated with low costs and few endoscopic interventions and can be placed in defects ranging from 5 mm up to about 2 cm, even in association with wound cavities and when internal drainage of purulent cavities is needed. EID has no rule in complete anastomotic dehiscence as double pigtails cannot be anchored.

Suturing techniques should only be applied by expert hands and in patients with fresh defects and aseptic wound conditions. New devices and methods such as the stent over sponge method or submucosal mesenchymal stem cell injections are on the way but larger clinical trials are needed in order to confirm the preliminary data.

Large-scale cohort studies comparing various treatment techniques for different defect sizes are missing and endoscopic guidelines are not available yet. Optimal treatment strategies according to defect type, size, anatomic characteristics and the presence of wound cavities need to be developed.

## Figures and Tables

**Figure 1 medicina-59-00136-f001:**
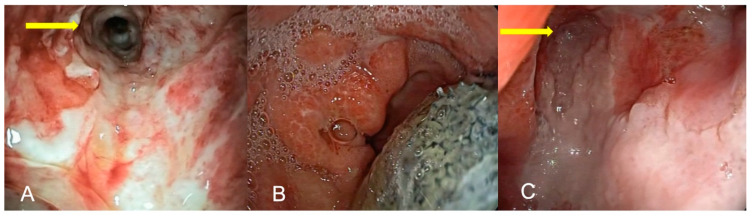
(**A**–**C**) partial closure of anastomotic esophageal defect after Ivor Lewis esophagectomy: (**A**) pleural drainage channel (yellow arrow); (**B**) EVT removal; (**C**) partial defect closure after EVT.

**Figure 2 medicina-59-00136-f002:**
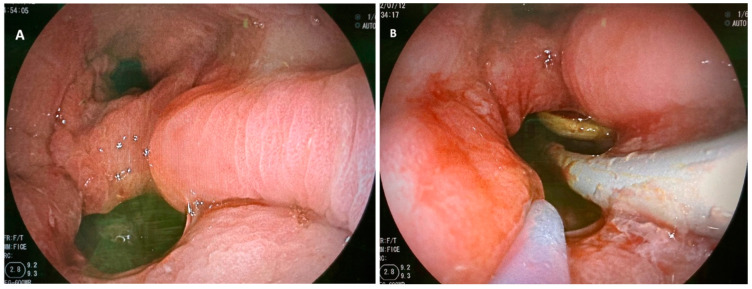
(**A**,**B**): Stapler line dehiscence after sleeve gastrectomy: (**A**) orifice and gastric inlet, (**B**) pigtail placement through the defect into the associated cavity.

**Figure 3 medicina-59-00136-f003:**
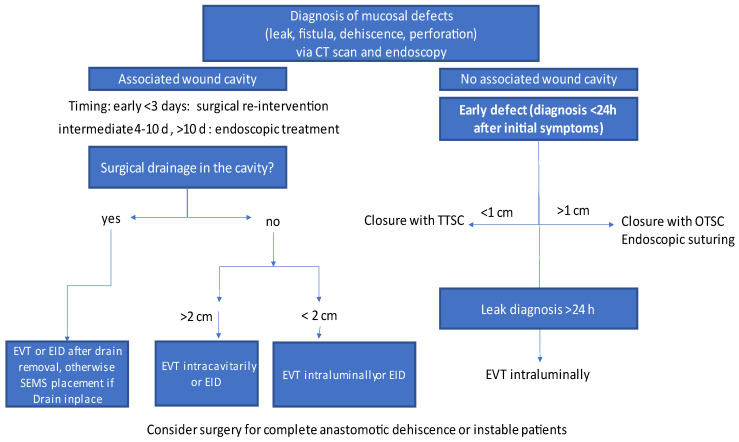
Flowchart treatment of mucosal defects in the upper GI tract, modified after Loschke and Di Leo et al. [15,103].

**Table 1 medicina-59-00136-t001:** Comparison of endoscopic closure techniques of upper GI fistula, leaks and perforations.

Device		Main Indication	Pros	Cons	AEs
STENT	SEPS FCSEMS PCSEMS BDs	Leaks Fistula Perforation > 2 cm	Easy placement High technical and clinical success Avoid stenosis Combined approach with clips	Expensive High migration rate Possible multiple sessions Need of percutaneous drainage of collection	Migration Food impaction Mucosal erosions Bleeding Perforation Stent ruptures Drooling, foreign body sensation
CLIP	TTSc	Leaks or perforations < 1 cm Acute perforations	Large availability Different shapes and sizes available Integration with other techniques	Limited efficacy Need of multiple interventions No full-thickness closure Need of percutaneous drainage of collection	Failure Migration
	OTSc	Leaks or perforations up to 2–3 cm Acute and chronic perforations	Full-thickness closure Single-step procedure	Need of percutaneous drainage of collection	Misdeployment
ENDOSUTURING	Overstitch Overstitch SX	Early defects > 2 cm	Full thickness closure High clinical and technical success	Expensive Need of high expertise Challenging use in angulated GI regions Need of percutaneous drainage of collection	Bleeding Strictures
ENDOSCOPIC VACUUM THERAPY (EVT)	Esosponge Suprasorb	Leaks, fistula, perforation with associated cavity	High clinical and technical success rate Simultaneous drainage of collection	Patient discomfort due to external tube drainage Need of multiple sessions	Bleeding Sponge ingrowth Strictures
ENDOSCOPIC INTERNAL DRAINAGE (EID)		Perforation and leaks with associated cavity	High clinical and technical success rate Low cost Oral feeding feasible	Need odultiple session	Bleeding Migration Splenic Hematoma

SEPS: self-expandable plastic stents; FCSEMS: fully covered self-expandable metallic stents; PCSEMS: partially covered self-expandable metallic stents; BDs: biodegradable stents; TTSc: through the scope clip; OTSc: over the scope clip.

## Data Availability

All articles listed in this review are listed in PubMed.

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
