# Peer review of "Endoscopic Management of Postoperative Esophageal and Upper GI Defects—A Narrative Review"

_medicina, 2023, doi:10.3390/medicina59010136_

Round 1

Reviewer 1 Report

The article “Endoscopic management of postoperative esophageal and upper GastroIntestinal defects” aims to answer an important question in the field. However, the guidelines for writing a systematic literature review are not mentioned (no methodology available) and overall the manuscript feels a little bit like a chapter in a book. Some other comments:

1.       The abstract should be revised. The authors start as if they are getting ready to present a large patient database/clinical study that could help address the controversies in the field. However, the manuscript is a review of available data

2.       Title is also misleading – the authors should specify that this is a review

3.       Why is this article novel?

4.       The section dedicated to stents would be greatly improved by a table summarizing all the data presented. As such, it is quite difficult to follow and the numerous abreviations make it even more difficult. Also, a comparison between stents underlying the benefits/disadvantages of each type of stent would be very helpful.

5.       The two sections dedicated to clips should be united. Similar to the previous section, I think a table would improve clarity. The same observation goes for most of the sections – I think clear, well-structured information with updated references would be most useful.

6.       How was the algorithm developed?  

English revisions required. Some (not all) examples are listed below:

Not least we proposed an algorithm of treatment considering the main variables that should be taken into account.”

„By now, endoscopic guidelines have not yet stated clear indications for the treatment of anastomotic defects after upper GI surgery”

„The use of endoluminal stent is a safe and effective treatment for upper GI leaks and fistula.”

„PCSEMSs has uncovered distal and proximal ends, that helps the stent to fit into its place (...)”

Author Response

Thank you very much for reviewing our manuscript and providing your useful comments. We tried to answer to your comments as sufficient as possible and changed the manuscript accordingly. Please find attached the response to every comment.

Reviewer 2 Report

Thank you for giving me an opportunity to review this review article. I think it is well-described about alternatives for upper GI defect after surgery. On the other hand, the authors should concisely provide tables of comparative studies among EVT, EID, and surgery. It is very important to build up their decision tree. 

1. Please provide a table of comparative studies among EVT, EID, and surgery.

2. Please provide tables of every alternatives you mentioned. TTSC, OTSC, EMS are major tools of endoscopic procedure. If possible, comparative studies should be also introduced in tables.

3. Discussion section is required. 

Author Response

(The authors gave the same response as above.)

Round 2

Reviewer 1 Report

thank you for addressing my comments

Reviewer 2 Report

The authors have revised their article as reviewers' suggestions.